# Clinical Features and Prognostic Factors for Primary Anaplastic Large Cell Lymphoma of the Central Nervous System: A Systematic Review

**DOI:** 10.3390/cancers13174358

**Published:** 2021-08-28

**Authors:** Yudai Hirano, Satoru Miyawaki, Shota Tanaka, Kazuki Taoka, Hiroki Hongo, Yu Teranishi, Hirokazu Takami, Shunsaku Takayanagi, Mineo Kurokawa, Nobuhito Saito

**Affiliations:** 1Department of Neurosurgery, The University of Tokyo, Bunkyo-ku, Tokyo 113-8655, Japan; yudai.hrn@gmail.com (Y.H.); stanaka@m.u-tokyo.ac.jp (S.T.); hongou-sin@umin.ac.jp (H.H.); yteranishi-nsu@umin.ac.jp (Y.T.); takami-tky@umin.ac.jp (H.T.); takayanagi-nsu@umin.ac.jp (S.T.); nsaito-nsu@m.u-tokyo.ac.jp (N.S.); 2Department of Hematology and Oncology, The University of Tokyo, Bunkyo-ku, Tokyo 113-8655, Japan; taokak-int@h.u-tokyo.ac.jp (K.T.); kurokawa@m.u-tokyo.ac.jp (M.K.)

**Keywords:** anaplastic large cell lymphoma, anaplastic lymphoma kinase, CD30, central nervous system, primary central nervous system lymphoma, prognostic factors

## Abstract

**Simple Summary:**

Primary anaplastic large cell lymphoma (ALCL) of the central nervous system (CNS) is a subtype of primary central nervous system lymphoma (PCNSL). ALCL is divided into anaplastic lymphoma kinase (ALK)-positive ALCL and ALK-negative ALCL, according to ALK expression. ALK-positive cancers tend to develop at a younger age and tend to have a better prognosis. Almost all past articles on primary ALCL of the CNS have been case reports and there have been no randomized trials or cohort studies on this subject. We thus performed a systematic review of primary ALCL of the CNS. According to the author’s survey, 36 case reports have been published in English-language journals. In this paper, we have summarized the clinical features and prognostic factors for primary ALCL of the CNS based on previous studies.

**Abstract:**

Primary anaplastic large cell lymphoma (ALCL) of the central nervous system (CNS) is a subtype of primary CNS lymphoma (PCNSL). There are very few comprehensive reports on this extremely rare tumor. Therefore, it is necessary to investigate the clinical features and prognostic factors for primary ALCL of the CNS. We performed a systematic review of the published literature. Past cases were comprehensively searched using PubMed, Cochrane Library, and Web of Science. Clinical information, such as age, sex, anaplastic lymphoma kinase (ALK) status, lesion sites, treatment methods, and survivorship were extracted. Thirty-nine cases with information on ALK status and treatment course were identified. The average observation period was 13 months, and the overall 2-year survival rate was 58%. Univariate analyses showed a statistically significantly better prognosis among patients < 40 years of age (*p* = 0.039, HR 0.32 (0.11–0.95)) and in relation to ALK positivity (*p* = 0.010, HR 0.24 (0.08–0.71) and methotrexate treatment (*p* = 0.003, HR 0.17 (0.05–0.56)). Because of the sparsity of cases, it is necessary to accumulate cases in order to perform more detailed analyses.

## 1. Introduction

Primary central nervous system lymphoma (PCNSL) is a rare subtype of extranodal non-Hodgkin lymphoma, which accounts for approximately 4% of central nervous system tumors [1]. Epidemiologically, the incidence of PCNSL is 0.43–0.47 per 100,000 person-years, with a statistically significantly higher incidence in men than in women [1]. Most PCNSL cases are B-cell lymphomas, and the most common histological type is diffuse large B-cell lymphoma (DLBCL) [2]. In contrast, primary central nervous system T-cell lymphoma accounts for approximately 2% of PCNSL cases [3]. Anaplastic large cell lymphoma (ALCL) is a T-cell or null cell lymphoma characterized by the proliferation of large lymphoid cells that express a large amount of CD30 on the cell surface; this subtype was first reported by Stein et al. in 1985 [4]. ALCL is divided into anaplastic lymphoma kinase (ALK)-positive and ALK-negative tumors according to the pattern of genetic abnormalities [5]. In ALK-positive ALCL, approximately 70–80% of cases have a specific translocation (t (2; 5) (p23q35)), which causes fusion of the nucleophosmin (NPM) nucleolar phosphoprotein gene and the ALK tyrosine kinase gene [6]. Primary ALCL of the CNS is a very rare histological type, and past articles have been limited to case reports. To clarify the clinical features of primary ALCL of the CNS and to explore factors related to prognosis, we conducted a systematic review of the current literature.

## 2. Materials and Methods

### 2.1. Study Selection

The study question was formulated using the PICOS (participants, interventions, comparisons, outcomes, and study design) strategy. The purpose of this study was to clarify the clinical features and factors associated with the prognosis for primary ALCL of the CNS. We conducted a systematic review in May 2021 using the PRISMA guidelines. This study has been registered in PROSPERO. The search was performed in the PubMed, Cochrane Library, and Web of Science databases using the following search keywords: “anaplastic large cell lymphoma” and “central nervous system or brain or meningeal”. The identified citations were screened by two authors (Y.H. and S.M.). The inclusion criteria were all case reports and case series of primary ALCL of the CNS. There have been no large-scale studies conducted on this subject because of the rarity of the disease. In addition, case reports and case series presenting cases with an unclear clinical course, without sufficient radiological imaging, or with an unclear ALK status, as well as non-English language publications, were excluded from the current review. A flow chart of the study selection process is presented in Figure 1. This analysis of previously published, anonymized data did not require an ethics review board approval or a formal exemption.

### 2.2. Data Extraction

Data extraction was independently performed by two independent investigators (Y.H. and S.M.). The data extracted included age, sex, ALK status, lesion sites, lesion multiplicity, surgical treatment type, treatment course (including chemotherapy and radiotherapy), survivorship, initial diagnosis, and initial treatment.

### 2.3. Statistical Analysis

To examine the prognostic factors, we performed an analysis based on data from past case reports reconsidering factors related to prognosis. We considered several factors: age, sex, ALK status, single or multiple lesions, meningeal lesions, type of surgery, methotrexate (MTX) therapy, and radiation therapy. The Fisher’s exact test was used to compare the frequency of each factor considered between the ALK-positive group and the ALK-negative group. The Wilcoxon rank sum test was used for the analysis of the age at diagnosis. Kaplan-Meier curves were drawn for each group, and *p*-values were calculated using the log-rank test to compare the survival distributions. We also performed a Cox regression analysis in order to determine the predictors of survivorship. All statistical analyses were performed using R (The R Foundation for Statistical Computing, Vienna, Austria).

## 3. Results

### 3.1. Clinical Features of Primary ALCL of the CNS

We identified two case series and 34 case reports in the current literature, comprising 39 cases of primary ALCL of the CNS with information on ALK status and treatment courses (Table 1) [7,8,9,10,11,12,13,14,15,16,17,18,19,20,21,22,23,24,25,26,27,28,29,30,31,32,33,34,35,36,37,38,39,40,41,42].

A summary of patient backgrounds and clinical features of the primary ALCL of the CNS cases included in the current review are described in Table 2; medical, demographic, and clinical information on the ALCL cases is presented in comparison to the literature for all PCNSL cases (Table 2).

The clinical features of primary anaplastic large cell lymphoma of the central nervous system have been summarized in Table 2. The median age was 21 years (range, 1–82 years), and the female-to-male ratio was 0.18. With regard to the imaging findings, 80% of cases presented with meningeal involvement and 38% of cases had multiple lesions. Cell markers were described in detail in 37 cases, 30 cases (81%) were T-cell type, and seven cases were null-cell type (19%). Regarding treatment methods, of the 33 cases that received surgical treatment, 16 were biopsies and 17 were tumor resections (including total or partial resection), and the ratio of biopsy and resection was approximately 50%. MTX-based chemotherapy was administered in 62% of the cases. The overall 2-year survival rate of ALCL was 58% (Figure 2), a poorer prognosis than that of general PCNSL (2-year survival rate: 90%) [43].

Of the 39 cases identified, 28 were ALK-positive and 11 were ALK-negative. The median age was 17.5 years in the ALK-positive group, while it was 63 years in the ALK-negative group. Of the 28 ALK-positive patients, 21 received MTX-based chemotherapy. On the other hand, of the 11 ALK-negative cases, only three patients had undergone MTX-based chemotherapy; the others received radiation treatment alone or best supportive care due to age and poor performance status. The ALK-positive group had a relatively good prognosis, with a 2-year survival rate of 71%. In contrast, the 2-year survival rate of the ALK-negative group was 22% and the median survival time was 0.21 years.

### 3.2. Factors Associated with the Prognosis for Primary ALCL of the CNS

As mentioned above, the average observation period for the 39 cases was 13 months. Univariate analyses via the log-rank test for each demographic and medical factor (age, sex, ALK status, lesion multiplicity, meningeal lesions, MTX chemotherapy, and radiation therapy) showed statistically significant differences in the prognosis for age < 40 years (*p* = 0.029), ALK-positive tumors (*p* = 0.005), and MTX treatment (*p* = 0.0009). Specifically, the 2-year survival rates for ALK-positive and ALK-negative cases were 71% and 22%, respectively (Figure 3).

A Cox regression analysis was performed to search for factors related to prognosis. As a result, age < 40 years (*p* = 0.039, HR 0.32 (0.11–0.95)), ALK-positive tumor (*p* = 0.010, HR 0.24 (0.08–0.71), and MTX-based chemotherapy (*p* = 0.003, HR 0.17 (0.05–0.56)) were factors associated with a good prognosis. No statistically significant differences were observed in terms of sex, single or multiple lesions, meningeal lesions, or radiation therapy (Table 3).

Since the majority of patients are in the ALK-positive group, a sub-analysis of only the ALK-positive group was performed in the same manner. As a result of subgroup analysis, there were no significant factors associated with a good prognosis. The results are summarized in Table 4.

In this study, given that most patients in the ALK-positive group were under 40 years of age, as well as the lack of chemotherapy for many patients in the ALK-negative group, it is not clear which factors are directly related to prognosis (Table 3). Multivariate analysis should be performed to eliminate the effects of confounding factors. However, in this study, we did not perform multivariate analysis due to the limited number of the samples.

## 4. Discussion

In this systematic review, the clinical features of primary ALCL of the CNS were clarified. We report a 2-year survival rate of 58% with a median follow-up of 13 months. In addition, age < 40 years, ALK-positive tumors and treatment with methotrexate were factors associated with good outcomes. To the best of our knowledge, this is the first systematic review and comprehensive analysis of the primary ALCL of CNS.

### 4.1. Patient Background

Differences in patient backgrounds between primary ALCL of the CNS and PCNSL include differences in age distribution and the proportion of immunosuppressed patients. The median age at diagnosis for general PCNSL was 66 years [44], while that of ALCL was 21 years. Human immunodeficiency virus/acquired immunodeficiency syndrome (HIV/AIDS) is a major risk factor for PCNSL [45], and PCNSL cases with HIV have different properties from those without. Approximately 19% of PCNSL patients have HIV in the United States [46]. On the other hand, for primary ALCL of the CNS, only one case was reported to be associated with AIDS [38]; all other cases were considered immunocompetent.

### 4.2. Diagnosis

It is very rare for primary ALCL of the CNS to be diagnosed correctly at an early stage. In fact, according to a previous report [42], an average of approximately 40 days is required for diagnosis; this suggests frequent delays until final diagnosis, and there have been reports of cases in which the delay in diagnosis became fatal. An awareness of the initial symptoms is important for accurate and early diagnosis; initial symptoms include headaches, seizures, ataxia, cognitive dysfunction, and hemiparesis. Regarding radiological features, lesions are present in the cerebral cortex, as well as in the brain stem and the spinal cord; the disease also involves the dura mater and sometimes presents with bone infiltration [12,32]. In this study, 80% of the cases presented with meningeal involvement, which suggests that ALCL originates in the dura. Small lesions along the dura are often initially diagnosed as inflammatory diseases, such as meningitis or sarcoidosis. The most commonly presumed causative infection is tuberculosis, which often causes leptomeningeal enhancement, as well as CNS lymphoma and sarcoidosis. Of the 33 confirmed cases receiving surgical treatment, 12 cases were diagnosed with infectious diseases, such as tuberculous, meningitis, and viral meningoencephalitis, and antibacterial, antiviral, and antituberculosis agents were started as an initial therapy. An ALCL mass is often diagnosed as a meningioma because of its attachment to the dura mater. Differential diagnoses for dural tumors include meningioma, hemangiopericytoma, metastatic brain tumors, and non-neoplastic lesions, such as sarcoidosis, tuberculosis, and IgG4-related disease [47]. Imaging features that distinguish meningioma mimics from meningiomas are the absence of a dural tail, homogenous T2 hypointensity or hyperintensity, osseous destruction, and leptomeningeal extension [48]. The uptake of fluorodeoxyglucose is increased on positron emission tomography scans for PCNSL. The mean maximum standard uptake value of PCNSL is approximately 15–25, which is higher than that of other tumors [49,50,51]. However, it is often difficult to properly diagnose ALCL based on the above-mentioned imaging features alone.

The accurate diagnosis of lymphoma requires tissue diagnosis. Prior to surgical treatment, a biopsy is recommended to provide a histological diagnosis when PCNSL is suspected [52]. It has long been reported that the degree of surgical removal of PCNSL does not affect the prognosis [53]. The usefulness of the total surgical removal of ALCL has not been proven. It has also been reported that resection may be considered if there is a single lesion and it is in a location that can be safely resected [54]. It may be useful to remove as much of the tumor as possible when symptoms of increased intracranial pressure are exhibited due to the mass effect of a large tumor. In fact, intracranial hypertension has been included as a cause of death due to ACLC of the CNS in one case report [9].

### 4.3. ALK Positivity

Regarding prognosis, previous reports have cited ALK-positive patients age <40 years, and chemotherapy with any regimen as good prognostic factors [11,21]. According to the World Health Organization (WHO) classification, ALCL is classified as ALK-positive and ALK-negative cases according to the expression of ALK [5]. In ALK-positive ALCL, approximately 70–80% of cases have a specific translocation (t (2; 5) (p23q35)), which causes fusion of the NPM nucleolar phosphoprotein gene and the ALK tyrosine kinase gene [6,55]. Constitutively activated NPM-ALK kinase acts as a trigger for many signaling pathways, leading to the malignant transformation of cells [56]. On the other hand, this tumor is ALK-negative in approximately 40–50% of ALCL cases [57], and various molecular biological factors are involved in ALK-negative cases. Gain-of-function of Janus kinase 1 (JAK1) and the signal transducer and activator of transcription 3 (STAT3) mutations cause constitutive activation of the JAK-STAT pathway, which has been identified in approximately 18% of ALK-negative ALCL cases [58]. Almost all of the above genetic profiles are found in reports of systemic ALCL. To our knowledge, there have been no reports of detailed genetic analyses for primary ALCL of the CNS. In the current analysis, we found that clinical features were different between the ALK-positive and-negative groups. Specifically, when stratified by ALK status, we found that all patients in the ALK-positive group were 40 years of age or younger, whereas all but one patient in the ALK-negative group were 45 years or older. Even with systemic ALCL, ALK-positive cases develop during the first 30 years of life, whereas ALK-negative cases develop mainly in patients aged 40–65 years [55]. Regarding the prognosis of primary ALCL of the CNS, the ALK-positive group had a statistically significantly better prognosis than the ALK-negative group. For systemic ALCL, the ALK-positive group has a better prognosis than the ALK-negative group, similar to primary ALCL of the CNS. The 5-year overall survival rates for systemic ALCL was 70–90% in patients with ALK-positive ALCL and 30–50% in patients with ALK-negative ALCL [55,59].

### 4.4. Chemotherapy

High-dose methotrexate at doses exceeding 3.5 g/m^2^ is one of the most important induction chemotherapies for PCNSL [60,61]. High-dose methotrexate chemotherapy was associated with a significant improvement in the 2-year overall survival compared with the no therapy group in this study (77% and 29%, respectively, *p* = 0.0009). This result suggests that high-dose methotrexate is an important chemotherapy for ALCL of CNS, as well as other PCNSL. The addition of high-dose cytarabine is advised in patients younger than 75 years of age [62]. Cyclophosphamide, doxorubicin, vincristine and prednisolone (CHOP) therapy, which is the standard treatment for systemic ALK-positive and-negative ALCL, was shown to be transiently responsive but rapidly resistant; this is partially due to the inadequate penetration of the blood–brain barrier [63]. Radiation therapy should be considered a part of consolidation [64]. However, radiation therapy is associated with increased neurocognitive deficits in elderly patients. There is no standard protocol for chemotherapy and radiotherapy for primary ALCL of the CNS. The mainstream treatment in this series was chemotherapy centered on methotrexate. Of the 28 ALK-positive patients, 21 received chemotherapy, including an MTX-based regimen. It has been reported in the past that CHOP therapy was ineffective for some PCNSL cases (prior to the year 2013); these cases are thought to have been treated according to the treatment protocol for systemic ALCL. Cases of MTX resistance have also been reported; cytarabine and etoposide (CYVE) therapy was effective in one case of small cell variant that recurred despite treatment with MTX [31]. While the prognosis for ALK-negative ALCL is poor, chemotherapy has not been administered in many cases. Eleven cases of ALK-negative ALCL were reported; only three underwent MTX-based chemotherapy, whereas the other cases received radiation alone or best supportive care due to age and poor performance status.

### 4.5. Prognosis

According to previous reports, ALK expression is not an independent prognostic factor in systematic ALCL because there is a correlation between ALK positivity and disease onset at a young age [65,66]. Since the number of patients in this cohort is small, it is not appropriate to perform multivariate analysis, and it is difficult to analyze whether ALK positivity is an independent factor. Performance status is a prognostic factor in PCNSL, thus, it is necessary to examine if the same can be said in primary ALCL of CNS. However, the information regarding this from past literatures is limited. Information on performance status in this study was also incomplete. It has been reported that CD56 is an independent prognostic factor for ALCL; CD56 expression in ALCL indicates a worse overall prognosis in both ALK-positive and ALK-negative subgroups [65]. Two CD56-positive cases have been reported within primary ALCL of the CNS in the past; and one case died within 1 month of the diagnosis [27,33]. It is currently unknown whether CD56 is a prognostic factor for primary ALCL of the CNS. The contents described in the discussion so far are itemized and summarized in the following table (Table 5).

### 4.6. Summary

This study conducted a systematic review of the rare disease primary ALCL of CNS, revealing the clinical features of the disease and prognostic factors. One of the strengths of this study is that we were able to clarify the clinical features by a comprehensive literature search. On the other hand, the major limitation is the small number of cases. At present, only 39 cases of primary ALCL of the CNS have been reported. It is desirable to analyze risk factors for each gender or perform multivariate analysis. Due to the very limited number of patients, it is difficult to perform the above-mentioned detailed analysis, which is expected for future research.

## 5. Conclusions

Primary ALCL of the CNS is extremely rare; a total of approximately 40 cases has been reported in the past. In this paper, we conducted a systematic review summarizing the clinical features and prognostic factors of the disease based on previous reports. Age <40 years, ALK-positive tumors and treatment with methotrexate were factors associated with good outcomes.

## Figures and Tables

**Figure 1 cancers-13-04358-f001:**
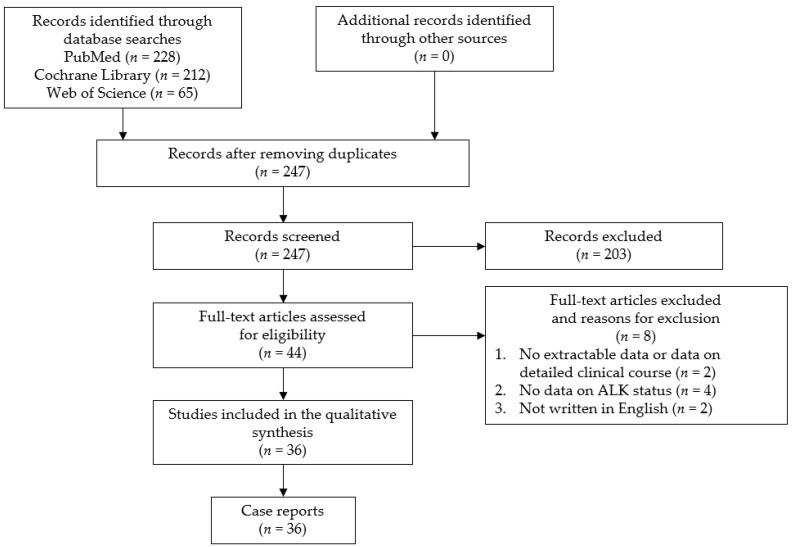
Flow chart of the study selection process. ALK: anaplastic lymphoma kinase.

**Figure 2 cancers-13-04358-f002:**
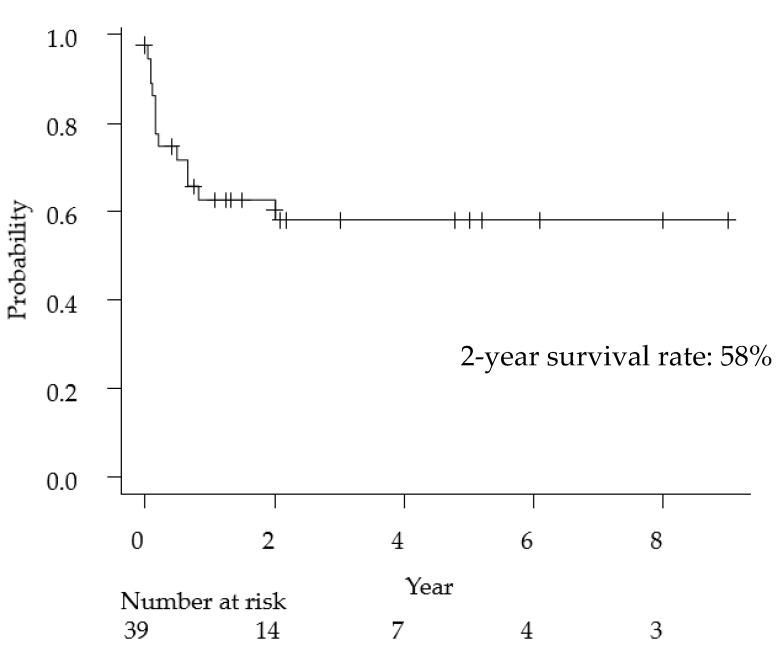
Kaplan-Meyer curve for overall survival following primary anaplastic large cell lymphoma of the central nervous system.

**Figure 3 cancers-13-04358-f003:**
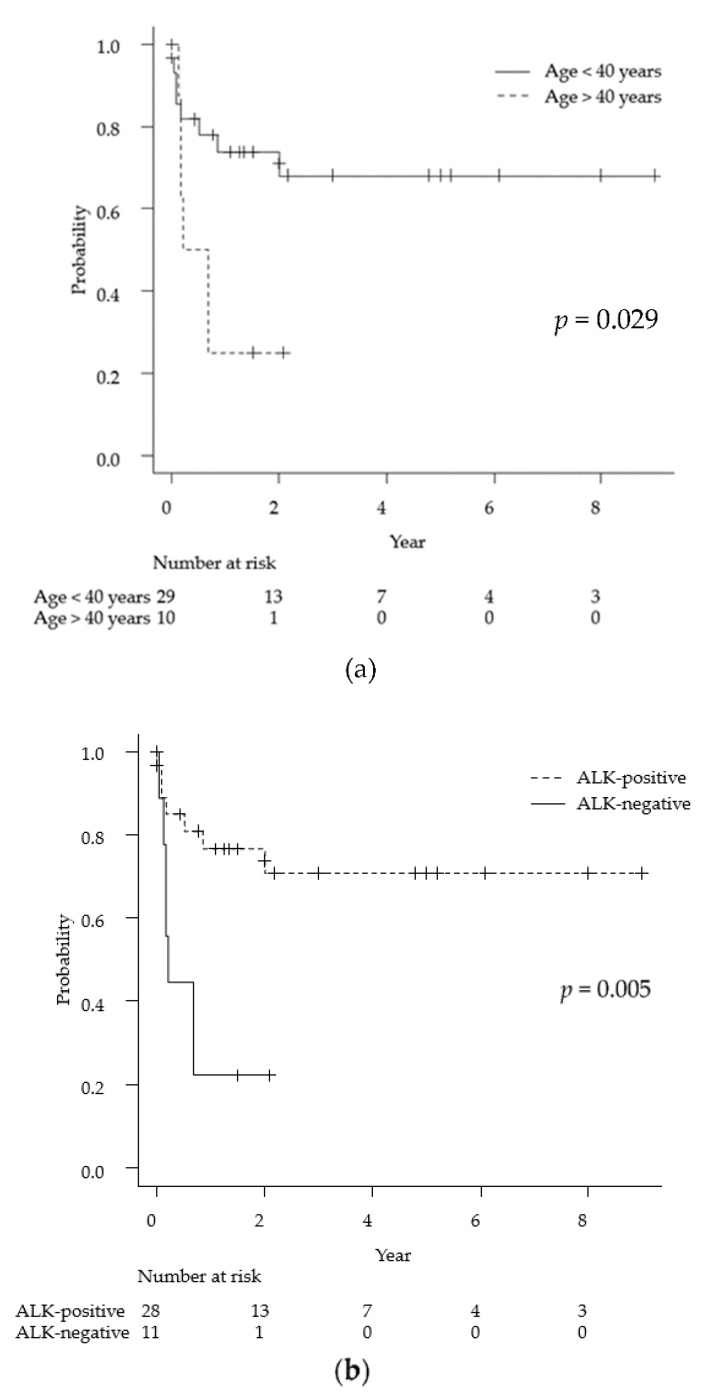
Kaplan-Meyer curves for overall survival for primary anaplastic large cell lymphoma of the central nervous system for each of the following factors: age (**a**), ALK positivity (**b**), and MTX treatment (**c**). ALK: anaplastic lymphoma kinase, MTX: methotrexate.

**Table 1 cancers-13-04358-t001:** List of reported cases of primary anaplastic large cell lymphoma of the central nervous system in the current literature.

Cases	Authors	Year	Age	Sex	ALK Status	Location	Lesion	Meningeal	Surgery	Marker	MTX	RT	Survival	Initial Symptom/s	Primary Diagnosis	First Therapy
1	Havlioglu et al. [7]	1995	4	F	positive	multifocal brain, brain stem, spinal cord	M	+	biopsy	null	-	+	NED at 6.1 years	headache, nausea, vomiting, neck stiffness	mycobacterial	anti-TB
2	Buxton et al. [8]	1998	10	F	positive	parietal lobe abutting against falx	S	+	resection	T	+	+	dead at 6 months following CT	sensory disturbance	ND	surgery
3	Abdulkader et al. [9]	1999	13	M	positive	frontal and parietal	M	+	biopsy	T	+	-	dead shortly after CT	headache, vomiting	mycobacterial	anti-TB
4	Ponzoni et al. [10]	2002	29	M	positive	frontal and temporal	M	+	biopsy	T	+	+	NED at 13 months	fever, headache, seizures	meningitis	antibiotics
5	George et al. [11]	2003	17	M	positive	parietal dura	S	+	ND	T	-	+	NED at 4.8 years	ND	ND	ND
6	George et al. [11]	2003	18	F	positive	temporal lobe, dura mater	M	+	ND	T	+	+	NED at 5.2 years	ND	ND	ND
7	Rupani et al. [12]	2005	17	M	positive	frontal and parietal lobe invading scapura	M	+	biopsy	T	-	+	dead at 1 month	headache, upper limb hemiparesis, partial seizures	mycobacterial	anti-TB
8	Cooper et al. [13]	2006	39	M	positive	occipital and parietal	S	-	biopsy	T	+	+	NED at 9 months	seizures	headaches	none
9	Carmichael et al. [14]	2007	38	M	positive	parietal, occipital	S	ND	biopsy	T	+	+	NED at 15 months	seizure, syncope, hemiparesis, visual field deficit, ataxia	glioblastoma or PCNSL	WBRT
10	Karikari et al. [15]	2007	4	M	positive	pineal, frontal, parietal	M	+	biopsy	T	-	+	NED after CT	seizures, altered mental status	meningitis	antibiotics
11	Merlin et al. [16]	2008	13	M	positive	frontal dura	S	+	biopsy	ND	+	+	dead at 2 years	headache, nausea	meningitis	CRT
12	Ozkaynak et al. [17]	2009	9	M	positive	bilateral frontal	M	ND	partial resection	T	+	+	NED at 26 months	altered mental status	meningitis	antibiotics
13	Shah et al. [18]	2010	1	M	positive	dura mater	S	+	partial resection	T	+	-	NED at 9 years	progressive lethargy, motor function loss.	ALCL	chemo
14	Vivekanandan et al. [19]	2011	20	M	positive	sylvian fissure	S	+	resection	T	-	+	NED at 8 years	seizures	ALCL	CRT
15	Kim et al. [20]	2013	30	M	positive	parietal and occipital dura	S	+	resection	T	+	+	NED at 16 months	headache	meningioma or PCNSL	surgery, CRT
16	Nomura et al. [21]	2013	20	M	positive	frontal	S	ND	resection	T	+	-	NED at 5 years	seizures	high grade glioma	surgery, chemo
17	Park et al. [22]	2013	31	M	positive	leptomeningeal	S	+	biopsy	T	+	-	NED at 18 months	altered mental status, headache	meningitis	antibiotics
18	Dunbar et al. [23]	2014	10	M	positive	frontal	S	+	biopsy	T	+	-	NED at 3 years	aphasia, hemiparesis	meningitis	antibiotics
19	Furuya et al. [24]	2014	11	M	positive	parietal	S	+	biopsy	null	+	+	NED at 8 years	headache, nausea	meningitis	antibiotics
20	Geetha et al. [25]	2014	19	M	positive	cerebellum	S	ND	partial resection	ND	+	-	dead at 10 months	headache, vomiting	tumor	surgery
21	Kuntegowdenahalli et al. [26]	2015	18	M	positive	parietal, occipital	S	+	resection	T	+	+	NED after CT	fever, headache, seizures	tumor	surgery, CRT
22	Dong et al. [27]	2016	34	M	positive	spinal cord	M	+	biopsy	null	+	+	NED at 2 years	headache, diplopia, vomiting	mycobacterial	anti-TB
23	Splavski et al. [28]	2016	26	M	positive	intraventricular	S	-	resection	T	+	+	NED at 2 years	diplopia, ptosis	tumor	surgery
24	Kaku et al. [29]	2017	21	M	positive	frontoparietal	S	+	resection	T	+	+	NED at 2 years	headache, seizures, fever	tumor	surgery
25	Feng et al. [30]	2020	8	M	positive	parietal	S	+	CSF	T	-	-	dead at 8 weeks	cognitive impairment, dizziness, vomiting, fever, convulsions	viral encephalitis	antibiotics
26	Hirano et al. [31]	2020	26	M	positive	occipital dura	S	+	resection	T	+	-	NED at 5 months	headache, blurry vision	meningioma	surgery
27	Lee et al. [32]	2020	12	M	positive	parietal	S	+	resection	null	+	-	NED at 16 months	headache, seizures	PCNSL or meningioma	surgery
28	Liu Q et al. [33]	2020	12	M	positive	occipital, falx	M	+	biopsy	null	-	-	dead at 1 month	headache, vomiting	tumor	none
29	Paulus et al. [34]	1994	63	M	negative	parietal and frontal	M	-	biopsy	T	-	+	dead at 11 weeks	ND	tumor	surgery
30	Chuang et al. [35]	2001	46	F	negative	parietooccipital	S	+	resection	T	-	+	NED at 25 months	headache, hemiparesis, blurry vision	tumor	surgery
31	George et al. [11]	2003	22	F	negative	dura cerebellum, temporal, 4 additional lesions	M	-	ND	T	-	-	dead at 11 days	ND	ND	ND
32	George et al. [11]	2003	50	M	negative	parietal, 2 additional supratentorial, dura	M	+	ND	null	-	+	dead at 2 months	ND	ND	ND
33	Gonzales et al. [36]	2003	82	M	negative	posterior fossa attaching to tentorium	S	+	ND	T	-	-	dead at 6 weeks	ND	ND	ND
34	Tajima et al. [37]	2003	52	F	negative	frontal	M	-	biopsy	null	+	+	NED (lesions markedly decreased)	hemiparesis	leukoenchephalopathy	biopsy
35	Rowsell et al. [38]	2004	46	M	negative	occipital	S	+	resection	T	-	+	dead at 2 months	ataxia	tumor	surgery
36	Kodama et al. [39]	2009	79	M	negative	parietoocipital	S	-	resection	T	-	-	dead at 8 months	dementia-like symptoms	tumor	surgery
37	Colen et al. [40]	2010	65	M	negative	temporal	S	+	resection	T	+	+	NED after 2 courses of CT	headache, blurry vision	meningioma	surgery
38	Sugino et al. [41]	2012	75	M	negative	bilateral hemisphere	M	-	biopsy	T	-	+	dead at 8 months	memory loss	tumor	biopsy
39	Lannon M et al. [42]	2020	63	M	negative	intracranial and intraspinal	M	+	resection	T	+	-	NED at 18 months	leg weakness	sarcoidosis or lymphoma	steroids

ALCL, anaplastic large cell lymphoma; ALK, anaplastic lymphoma kinase; CRT, chemoradiotherapy; CSF, cerebrospinal fluid; CT, chemotherapy; F, female; M, male; M, multiple; MTX, methotrexate; NED, no evidence of disease; ND, no data; S, single; PCNSL, primary central nervous system lymphoma; RT, radiation therapy; T, T-cell type; TB, tuberculosis; WBRT, whole brain radiotherapy.

**Table 2 cancers-13-04358-t002:** Summary of the clinical features of primary anaplastic large cell lymphoma of the central nervous system cases based on past reports, including in comparison to primary central nervous system lymphoma cases.

Variables	Primary ALCL of the CNS	Primary ALK-Positive ALCL of the CNS	Primary ALK-Negative ALCL of the CNS	*p*-Value
Incidence	39 cases reported	28 cases reported	11 cases reported	-
Median age	21 (1–82)	17.5 (1–39)	63 (22–82)	<0.01
Female to male ratio	0.18	0.12	0.38	0.324
Multiple lesions	15 (38%)	9 (32%)	6 (55%)	0.277
Meningeal involvement	31 (80%)	25 (92%)	6 (55%)	0.028
Immunodeficiency	1 (3%)	0 (0%)	1 (9%)	0.282
MTX-based chemotherapy	24 (62%)	21 (75%)	3 (27%)	0.010
Radiation therapy	25 (64%)	18 (64%)	7 (64%)	1.000
Prognosis	58% (2-year survival)	71% (2-year survival)	22% (2-year survival)	-
Median survival time	Not reached	Not reached	0.21 years	-

ALCL, anaplastic large cell lymphoma; ALK, anaplastic lymphoma kinase; CNS, central nervous system; MTX, methotrexate.

**Table 3 cancers-13-04358-t003:** Factors related to prognosis for primary anaplastic large cell lymphoma of the central nervous system.

Variables	HR (95% CI)	*p*-Value
Age < 40 years	0.32 (0.11–0.95)	0.039
Female sex	1.00 (0.22–4.49)	0.998
ALK positive	0.24 (0.08–0.71)	0.010
Multiple lesions	2.23 (0.78–6.37)	0.135
Meningeal lesions	0.51 (0.16–1.67)	0.265
Tumor resection *	0.47 (0.14–1.60)	0.226
MTX-based chemotherapy	0.17 (0.05–0.56)	0.003
Radiotherapy	0.49 (0.17–1.41)	0.189

ALK, anaplastic lymphoma kinase; CI, confidence interval; HR, hazard ratio; MTX, methotrexate. * includes total or partial resection of the lesions, but not biopsy.

**Table 4 cancers-13-04358-t004:** Factors related to prognosis for ALK-positive group of primary anaplastic large cell lymphoma of the central nervous system.

Variables	HR (95% CI)	*p*-Value
Female sex	1.16 (0.14–9.67)	0.891
Multiple lesions	2.01 (0.45–9.04)	0.362
Meningeal lesions	N/A	N/A
Tumor resection *	0.39 (0.08–2.01)	0.260
MTX-based chemotherapy	0.34 (0.07–1.53)	0.160
Radiotherapy	0.35 (0.08–1.59)	0.175

ALK, anaplastic lymphoma kinase; CI, confidence interval; HR, hazard ratio; MTX, methotrexate; N/A, not available. * includes total or partial resection of the lesions, but not biopsy.

**Table 5 cancers-13-04358-t005:** Summary of clinically important features for primary anaplastic large cell lymphoma of the central nervous system.

**Patient Background** Median age: 21 years Usually not related to AIDS
**Diagnosis** Difficulty in accurate early diagnosis Involving the dura Tend to be misdiagnosed as meningioma, meningitis, tuberculosis, sarcoidosis, etc. MRI findings of meningioma mimics: absence of a dural tail, homogenous T2 hypointensity or hyperintensity, osseous destruction, and leptomeningeal extension FDG-PET: higher maximum standard uptake value (15–25)
**Treatment** Purpose of surgery: diagnosis, release of mass effect Induction chemotherapy: high-dose MTX Cytarabine and etoposide therapy is also considered in cases of MTX resistance
**Prognosis** Factors associated with good prognosis: age <40 years, ALK-positive tumors and treatment with methotrexate

AIDS, acquired immunodeficiency syndrome; ALK, anaplastic lymphoma kinase; FDG-PET, fluorodeoxyglucose-position emission tomography; MRI, magnetic resonance imaging; MTX, methotrexate.

## Data Availability

Data used for this study, though not available in a public repository, will be made available to other researchers upon reasonable request.

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
