# Peer review of "Clinical Features and Prognostic Factors for Primary Anaplastic Large Cell Lymphoma of the Central Nervous System: A Systematic Review"

_cancers, 2021, doi:10.3390/cancers13174358_

Round 1
Reviewer 1 Report
Summary
The manuscript (title: Clinical features and prognostic factors for primary anaplastic large cell lymphoma of the central nervous system: a systematic review) studied the relationship between the clinical features or factors and the prognosis for primary Anaplastic large cell lymphoma (ALCL) of the central nervous system (CNS). The primary Anaplastic large cell lymphoma of the central nervous system is a rare disease, the authors made great efforts to collected and cleaned the data from several different data sources. I have only one minor comment on the manuscript.
Comment:
- The collected data showed that the female-to-male ratio was 0.18. It implied that the claims and conclusion in the manuscript is applicable to men, probably not applicable to women. I suggest the authors point out it.
Author Response
Reviewer #1
- The collected data showed that the female-to-male ratio was 0.18. It implied that the claims and conclusion in the manuscript is applicable to men, probably not applicable to women. I suggest the authors point out it.
Response:
Thank you for your valuable feedback. Indeed, as you pointed out, the proportion of male patients in this cohort was so high that we need to consider the risk factors for each gender. However, due to the small number of patients overall, it was difficult to analyze risk factors by gender. I mentioned that in the discussion section.
Line 349
It is desirable to analyze risk factors for each gender or perform multivariate analysis. Due to the very limited number of patients, it is difficult to perform the above-mentioned detailed analysis, which is expected for future research.
Reviewer 2 Report
The authors have produced an original work regarding CNS-ALCL, by reviewing litterature cases.
The article is well written and, overall, of interest for the reader as those cases are trully rare/exceptionnals:
- The References are in number in the text, but those number don't show up in the bibliography, and the reviewer has not been able to check references, which is a blocking point for further analysis.
Some remarks may enhance the manuscript:
- Regarding the methods,
- A ChiSq test should be avoided and we'd prefer a fisher test, as most of the comparisons involve a very limited number of cases (Yate's continuity correction for n<5).
- Multivariate survival analysis shouldn't have been conducted considering the very limited set of patient, although the reasons to conduct such a test are good. All the covariates share a lot of colinearity regarding ALK status (aknowlededged line 153-155), thus MV should have included interaction factor, but once again, because of the number of patients it is not feasible and shouldn't be carried as it is false.
- A sub analysis of ALK+ patients only is deemed, as this is the most numerous cohort, without the age/mtx issues.
- Table 2 is detailled and sel-sufficient, it should not be entirely detailled in the text (= should be shortened)
- Discussion is expanding behond the subject (MRI findings), and although the discussed points are of interest, authors may try to condensate their discussion in a short bullet-point table for diagnosis highlights, treatments, etc...
Author Response
Reviewer #2
- The References are in number in the text, but those number don't show up in the bibliography, and the reviewer has not been able to check references, which is a blocking point for further analysis.
Response:
Thank you for your valuable comments. As you pointed out, we have numbered the references in the references section.
- Regarding the methods, A ChiSq test should be avoided and we'd prefer a fisher test, as most of the comparisons involve a very limited number of cases (Yate's continuity correction for n<5).
Response:
Thank you for your valuable feedback. As you pointed out, it is better to use a fisher test instead of a chi-square test for statistical analysis. It was reanalyzed and reflected in Table 2.
Line 83
The fisher exact test was used to compare the frequency of each factor considered between the ALK-positive group and the ALK-negative group.
- Multivariate survival analysis shouldn't have been conducted considering the very limited set of patient, although the reasons to conduct such a test are good. All the covariates share a lot of colinearity regarding ALK status (aknowlededged line 153-155), thus MV should have included interaction factor, but once again, because of the number of patients it is not feasible and shouldn't be carried as it is false.
Response:
Thank you for your valuable comments. As you pointed out, it is not appropriate to perform multivariate analysis with a very small number of set of patients. All the items related to multivariate analysis were deleted, and the reason why the analysis was not performed was clarified.
Line 197
Multivariate analysis should be performed to eliminate the effects of confounding factors. However, in this study, we did not perform multivariate analysis due to the limited number of the samples. ,
Line 320
Since the number of patients in this cohort is small, it is not appropriate to perform multivariate analysis, and it is difficult to analyze whether ALK positivity is an inde-pendent factor.
- A sub analysis of ALK+ patients only is deemed, as this is the most numerous cohort, without the age/mtx issues.
Response:
Thank you for your valuable comments. In this cohort, ALK-positive patients accounted for the majority. As recommended, we performed a sub-analysis on this and added the results to the text and table.
Line 183
Since the majority of patients are in the ALK-positive group, a sub-analysis of only the ALK-positive group was performed in the same manner. As a result of subgroup analysis, there were no significant factors associated with good prognosis. The results are summarized in Table 4.
Table 4. Factors related to prognosis for ALK-positive group of primary anaplastic large cell lymphoma of the central nervous system.
|
|
HR (95% CI) |
P-value |
|
Female sex |
1.16 (0.14-9.67) |
0.891 |
|
Multiple lesions |
2.01 (0.45-9.04) |
0.362 |
|
Meningeal lesions |
N/A |
N/A |
|
Tumor resection* |
0.39 (0.08-2.01) |
0.260 |
|
MTX-based chemotherapy |
0.34 (0.07-1.53) |
0.160 |
|
Radiotherapy |
0.35 (0.08-1.59) |
0.175 |
ALK, anaplastic lymphoma kinase; CI, confidence interval; HR, hazard ratio; MTX, methotrexate; N/A, not available. *includes total or partial resection of the lesions, but not biopsy.
- Table 2 is detailled and sel-sufficient, it should not be entirely detailled in the text (= should be shortened)
Response:
Thank you for your valuable comments. As you pointed out, the description of the contents of Table 2 has been shortened.
Line 135
The median age was 17.5 years in the ALK-positive group while it was 63 years in the ALK-negative group. Of the 28 ALK-positive patients, 21 received MTX-based chemo-therapy. On the other hand, of the 11 ALK-negative cases, only three patients had un-dergone MTX-based chemotherapy;
- Discussion is expanding behond the subject (MRI findings), and although the discussed points are of interest, authors may try to condensate their discussion in a short bullet-point table for diagnosis highlights, treatments, etc...
Response:
Thank you for your valuable comments. As you pointed out, we created a bulleted table about the contents of the discussion (Table 5).
Line 330
The contents described in the discussion so far are itemized and summarized in a table (Table 5). Table 5. Summary of clinically important features for primary anaplastic large cell lymphoma of the central nervous system.
|
Patient Background l Median age: 21 years l Usually not related to AIDS |
|
Diagnosis l Difficulty in accurate early diagnosis l Involving the dura l Tend to be misdiagnosed as meningioma, meningitis, tuberculosis, sarcoidosis, etc. l MRI findings of meningioma mimics: absence of a dural tail, homogenous T2 hypointensity or hyperintensity, osseous destruction, and leptomeningeal extension l FDG-PET: higher maximum standard uptake value (15-25) |
|
Treatment l Purpose of surgery: diagnosis, release of mass effect l Induction chemotherapy: high-dose MTX l Cytarabine and etoposide therapy is also considered in cases of MTX resistance |
|
Prognosis l Factors associated with good prognosis: age <40 years, ALK-positive tumors and treatment with methotrexate |
AIDS, acquired immunodeficiency syndrome; ALK, anaplastic lymphoma kinase; FDG-PET, fluorodeoxyglucose-position emission tomography; MRI, magnetic resonance imaging; MTX, methotrexate.